# Potential Use of Biochar in Pit Latrines as a Faecal Sludge Management Strategy to Reduce Water Resource Contaminations: A Review

Matthew Mamera [1,*], Johan J. van Tol [1], Makhosazana P. Aghoghovwia [1], Alfredo B. J. C. Nhantumbo [2], Lydia M. Chabala [3], Armindo Cambule [2], Hendrix Chalwe [3], Jeronimo C. Mufume [4] and Rogerio B. A. Rafael [2]

[1] Department of Soil, Crop and Climate Sciences, Faculty of Natural Sciences, University of the Free State, Bloemfontein 9301, South Africa; vantoljj@ufs.ac.za (J.J.v.T.); AghoghovwiaMP@ufs.ac.za (M.P.A.)

[2] Rural Engineering Department, Faculty of Agronomy and Forest Engineering, Eduardo Mondlane University, P.O. Box 257, Maputo 1102, Mozambique; abnhantumbo@yahoo.com (A.B.J.C.N.); armindo.cambule@uem.mz (A.C.); rogerborguete@gmail.com (R.B.A.R.)

[3] Department of Soil Science, School of Agricultural Sciences, University of Zambia, P.O. Box 32379, Lusaka 10101, Zambia; lchabala@unza.zm (L.M.C.); hendrix.chalwe@unza.zm (H.C.)

[4] Department of Morphological Sciences, Faculty of Medicine, Eduardo Mondlane University, P.O. Box 257, Maputo 1102, Mozambique; jmufume@gmail.com

\* Correspondence: 2015319532@ufs4life.ac.za; Tel.: +27-67-090-4564

**Abstract:** Faecal sludge management (FSM) in most developing countries is still insufficient. Sanitation challenges within the sub-Saharan region have led to recurring epidemics of water- and sanitation-related diseases. The use of pit latrines has been recognised as an option for on-site sanitation purposes. However, there is also concern that pit latrine leachates may cause harm to human and ecological health. Integrated approaches for improved access to water and sanitation through proper faecal sludge management are needed to address these issues. Biochar a carbon-rich adsorbent produced from any organic biomass when integrated with soil can potentially reduce contamination. The incorporation of biochar in FSM studies has numerous benefits in the control of prospective contaminants (i.e., heavy metals and inorganic and organic pollutants). This review paper evaluated the potential use of biochar in FSM. It was shown from the reviewed articles that biochar is a viable option for faecal sludge management because of its ability to bind contaminants. Challenges and possible sustainable ways to incorporate biochar in pit latrine sludge management were also illustrated. Biochar use as a low-cost adsorbent in wastewater contaminant mitigation can improve the quality of water resources. Biochar-amended sludge can also be repurposed as a useful economical by-product.

**Keywords:** biochar; contaminants; pit latrines; sludge management; sustainable soil conditioner; water quality

## 1. Introduction

Faecal sludge management (FSM) in most developing countries of the sub-Saharan region is ineffective and insufficient, which causes a deepening of sanitation problems [1–4]. Improper pit emptying and sludge disposal have been attributed to factors such as shortages in suitable sanitation, poor drainage systems, and high groundwater fluctuations [1,5]. Further, sludge management is impacted by high transport and disposal costs in landfills. The permanent airspace disposals can also lead to human and environmental impacts [6]. Previous and recent latrine building projects have focused on constructing latrines without considering the emptying process and sludge management strategies [2].

Sanitation challenges within sub-Saharan Africa have led to recurring epidemics of sanitation-related diseases, including soil-transmitted helminth infections [7]. Outbreaks can occur periodically where water supplies and sanitation provisions are inadequate, most

frequently in the developing world [3,4,8,9]. Between 1970 and 2011, African countries reported over 3 million suspected cholera cases, representing 46% of all cases reported globally [8]. Sub-Saharan Africa accounted for 86% of reported cases and 99% of deaths worldwide in 2011 [9–11]. Statistics of this nature are alarming and need urgent redress. While the reasons for these conditions are complex, part of the problem is the difficulty in accessing clean water and safe potable water, lack of sanitation, and the high costs involved. Pollution problems from pit latrines depend on climatic conditions, geological formations and soilscapes on the rate of soil contaminants migration. These factors lead to a need for scientific assessment of sludge management and pollution challenges. This can ensure that these sanitations are properly sited, designed, installed, monitored, and maintained [3,12]. Although the use of pit latrines as compared to open defecation can be beneficial, there are still concerns that they may cause dreadful human and ecological health impacts. This is associated with microbiological and chemical contamination of drinking water supplies through leaching into groundwater and surface water [11].

Integrated approaches for access and improvement of sanitation and water are needed to address these issues to curb the potential danger to public health and the environment. Creating simple and sustainable solutions for managing human excreta plays a direct role in slowing down the rate of environmental damage. This can be done by seeking alternative means that aim at reducing environmental pollution by faecal sludge, while not further depleting severely limited freshwater resources. Incorporation of soil a conditioner such as biochar has a high impact on the reduction in contaminant leaching [13–17].

Biochar is a high carbon-rich adsorbent produced from any organic biomass at high temperatures in conditions with limited oxygen [18]. Many studies to date have mostly focused on the potential of biochar to improve soil fertility for agricultural uses [19–21]. There are, however, numerous prospective benefits of integrating biochar in FSM studies. Such merits include: micro-organic mitigations [22]; reducing malodour [23]; contaminant barrier (bacteria and heavy metals) [15,24,25]; reduction in nitrogen [26], and carbon dioxide losses [27].

A gap in knowledge regarding the use of biochar to reduce the environmental threat of faecal sludge still exists. In the recent past, the potential of biochar to reduce leaching has been recognized, and several studies have been conducted on organic and inorganic pollution restriction by biochar. This review aimed to evaluate the potential of biochar in FSM through literature, which focused on biochar, sanitation, and faecal sludge studies. This review merits attention, because it explores alternative means for faecal sludge management, which can also be implemented in developing countries such as Mozambique, South Africa, and Zambia to minimize seepage of pit latrine waters and provide a sustainable soil conditioner for crop production.

## 2. On-Site Sanitation Systems

On-site sanitation is characterized by treatment and disposal of human waste, which is not removed to an off-site sanitation system [28,29]. Such sanitation facilities store wastes at the site of disposal, which decompose in situ [30]. These systems have two main categories; the wet latrines, which use water for flushing, and the dry latrines, which function without water sources. The different types of on-site sanitation systems [28,30,31] are pit latrines, ventilated improved pit latrines (VIPs), urine diversion (UD) toilets (Figure 1), ecological sanitation (EcoSan) latrines, Fossa Alterna, anaerobic biogas reactors, and septic tanks. A common pit latrine is composed of a simple top structure constructed over a pit and collects waste [32]. Improved pit latrines are a simple and low-cost type of sanitation system [13].

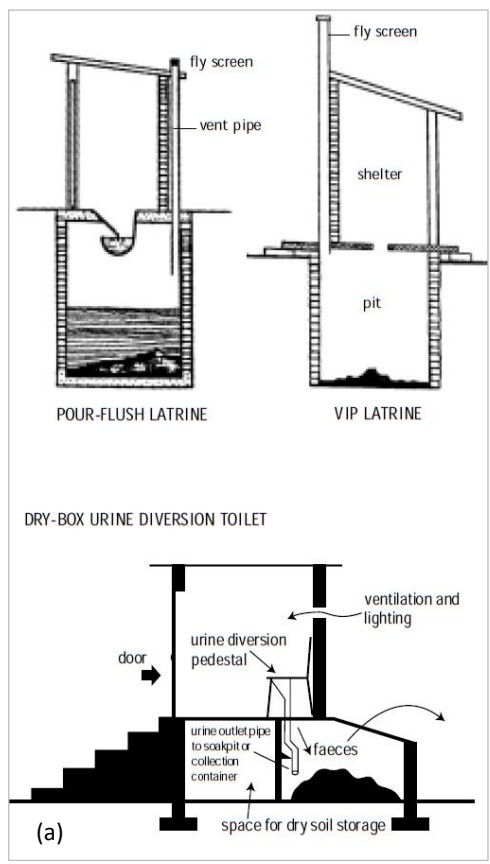

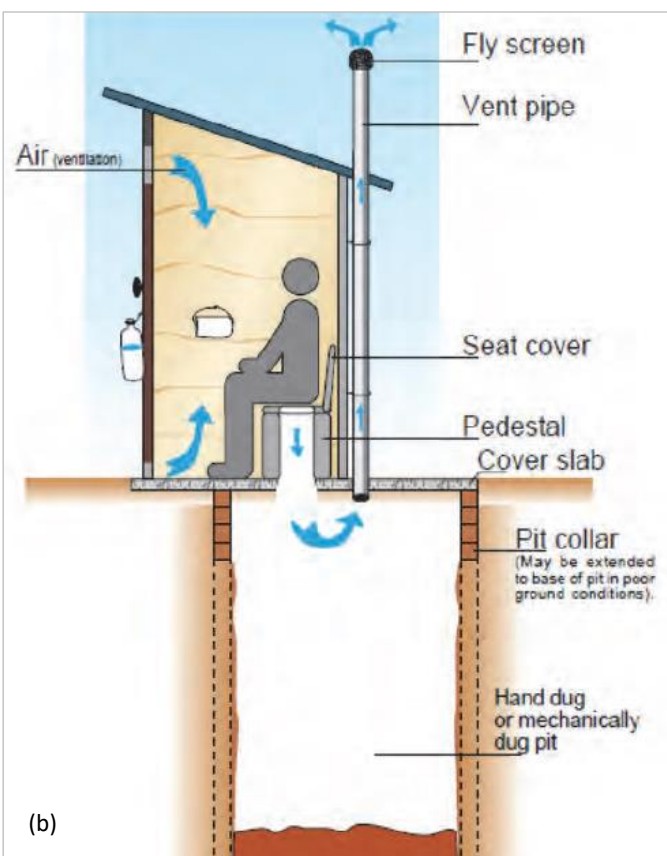

**Figure 1.** (**a**) Examples of on-site sanitation designs [30]; (**b**) typical structure for a VIP toilet system in South Africa [31] reproduced from the reference, copyright 2001, CC-BY-4.0.

### 2.1. Contamination Risks of Pit Latrines

On-site sanitation systems often represent a significant contamination threat towards groundwater associated with faecal matter accumulations, which can result in leaching of contaminants into the subsurface aquifer. Leachates in pits can lead to both microbiological and chemical contamination. In a pit latrine, the liquid fraction of waste that infiltrates into the soil is referred to as the hydraulic load [33]. Since pit latrines are usually not sealed [30], higher hydraulic loads can exceed natural attenuation potential in the sub-surface layers and cause direct contamination of groundwater sources. Designs of most pit latrines allow the liquid waste to infiltrate into the soil. Such wastes often contain micro-organisms and high nitrogen concentrations [30]. The hydrogeology in unlined pit latrines is extremely permeable, especially in coarser materials and fractured substratum. Such conditions promote rapid drainage in most natural soils [34]. Such designs of pit latrines allow for groundwater and surface water movements, which cause them to fill up rapidly [1,2]. Soil effluent infiltration rates of different soils not amended with carbon-based adsorbents such as biochar are shown in Table 1.

**Table 1.** Infiltration capacity of different soil types [33,35].

| Type of Soil | Infiltration Capacity Settled Sewage (L per $m^2$ per Day) |
|---|---|
| Coarse or medium sand | 50 |
| Fine sand, loamy sand | 33 |
| Sandy loam, loam | 25 |
| Porous silty clay and porous clay loam | 20 |
| Compact silty loam, compact silty clay, loam and non-expansive clay | 10 |
| Expansive clay | <10 |

Pour-flush latrines have a much greater hydraulic load as compared to dry latrines; thus they have a higher contamination capacity [35]. Pit latrines normally are deeper than other on-site sanitations and tend to rely on infiltration of leachate through the surrounding soil [30]. Pit latrines pose a contamination risk to water sources such as wells nearby. Therefore, wells need to be well covered. Kiptum and Ndambuki [36] found a strong correlation between the types of well cover, with the one made of concrete being better than the one made of timber. Concrete covers guard the well against surface runoff and windblown substances and help to exclude spilled water.

### 2.2. On-Site Sanitation Waste Components and Health Risks

Human excreta are composed of several chemicals (Table 2) and pathogens (Table 3) species posing threats to human health and the natural environment. Nitrates and phosphates are a major concern. Higher concentrations of nitrates (>45 mg/L) in drinking water sources are harmful to humans [37–39]. One of the effects of human beings ingesting water with high concentrations of nitrates is methemoglobinemia or infantile cyanosis, i.e., "blue baby syndrome" in infants and oesophageal cancer in adults [40]. The probable long-term effects of nitrate pollutants should be included in the planning phase of sanitation programmes, as remedial action is challenging, and blending with low nitrate waters may be the only viable option [41]. High loading of phosphates in water sources results in eutrophication problems, having an impact on human well-being, social interaction, economic activities, and the natural environment [42].

The majority of studies that assessed microbiological quality of groundwater in relation to pit latrines used faecal indicator bacteria, i.e., total coliforms, faecal coliforms, and *E. coli* [8,43]. Bacterial pathogens cause some of the best known and most feared infectious diseases, such as cholera, typhoid, and dysentery, which still cause massive outbreaks of diarrhoeal disease and contribute to ongoing infections [44]. Their control in drinking water remains critical in all countries worldwide [43].

**Table 2.** Human waste composition [42,45,46].

| Compound | Faeces (% Dry Weight) | Urine (% Weight) |
|---|---|---|
| Organic matter | 88–97 | 65–85 |
| Carbon (C) | 44–55 | 11–17 |
| Nitrogen (N) | 5.0–7.0 | 15–19 |
| Phosphorous ($P_2O_5$) | 3.5–4.0 | 2.5–5.0 |
| Potassium ($K_2O$) | 1.0–2.5 | 3.0–4.5 |
| Calcium (CaO) | 4.5 | 4.5–6.0 |
| Dry solids/person/day (g) | 30–70 | 50–70 |

**Table 3.** Common bacteria and viruses found in human excreta as pathogenic contaminants [33].

| Pathogen | Illness | Present in (Faeces/Urine) |
|---|---|---|
| *Escherichia coli*, *Faecal coliforms* | Diarrhoea | Both |
| *Leptospira interrogans* | Leptospirosis | Urine |
| *Salmonella typhi* | Typhoid | Both |
| *Shigella* spp. | Shigellosis | Faeces |
| *Vibrio cholerae* | Cholera | Faeces |
| Poliovirus | Poliomyelitis | Faeces |
| Rotaviruses | Enteritis | Faeces |

### 2.3. Heavy Metal Composition of Faecal Sludge

The disposal of heavy metals remains as a major concern globally to water sources contamination [47]. Heavy metals in faecal effluent originate from natural and anthropogenic sources [48]. A substantial quantity of the anthropogenic releases of heavy metals accumulates in surface and groundwater ecosystems [49]. Industrial water treatment plant (IWTP) sludge has higher concentration of heavy metals as compared to other sources such as water treatment plants (WTP) and wastewater treatment plants (WWTP). Thus, they are mostly not recommended for soil amendment and ecological purposes [50,51]. Several industrial sectors contribute heavy metals to the environment through sludge disposals. Some of these sources include plants such as galvanic processes, dye productions, steel pickling, electroplating industry, and the recycling of lead batteries, among many others [50]. Heavy metals concentration in pit latrines is lower than reported in wastewater sludge [52]. However, heavy metal elements are one of the main persistent contaminants of pit latrine leaching or municipal wastewater [48,53]. The persistence of heavy metals in effluent is caused by their non-biodegradable and harmful nature [54]. Metals are mobilized and transported into the food web because of the leaching process from waste dumps, polluted soils, and water [55]. The most common toxic heavy metals in wastewater and sewage sludge include arsenic (As), lead (Pb), mercury (Hg), cadmium (Ca), chromium (Cr), copper (Cu), nickel (Ni), silver (Ag), and zinc (Zn) [48,53,56–59]. There is increasing evidence linking Hg, Pb, As, and Cd toxicants to the incidence of cognitive impairments and cancers in children [60]. Additionally, high concentrations of arsenic and other heavy metals can affect the nervous system and kidneys and may cause reproductive disorders, skin lesions, endocrinal damage, and vascular diseases [8,37].

### 2.4. Treatment of Faecal Sludge

In some developing countries still relying on pit latrines, filled up latrines are either closed or emptied and the sludge disposed off-site as waste [17]. Sludge can also be utilized as a soil ameliorant, for improving the soil status [61]. However, land application of sludge can also promote the pollution of water and soil by heavy metals [62]. Prior to sludge applications, conventional treatments are carried out [62], but that is not normally the case in most developing countries. The removal of heavy metal pollutants can be achieved through these conventional techniques to treat wastewater streams, including reduction or precipitation via chemical means, ion exchange, electro-chemical methods, and reverse

osmosis. Nonetheless, such processes can be inadequate, especially for solutions with 1 to 100 (mg/L) of metal concentrations [63]. Other methods have also been successfully used for heavy metal removals, microbial remediation, and phytoremediation: cortex fruit wastes, including banana, kiwi, and tangerine peels [55]; activated carbon, peanut husk charcoal, fly ash, and natural zeolite [47]; composting and immobilization using biochar [64]. Such processes are cost effective, with non-hazardous end products [65]. The effective elimination of heavy metals from wastewater relies on several aspects, such as sludge concentration, the solubility of metal ions, pH, the metallic species and its concentration, and wastewater contamination load [63,66].

## 3. Biochar Adsorbents

Biochar is a material that has only recently been studied as an environmental amendment [16,67,68]. The use of biochar in pit latrine sludge treatment in most developing countries is still limited. This is primarily due to a lack of awareness in communities relying on pit latrines on contaminant immobilization potential of biochar [17,29]. Biochar applications historically predate several years in the Brazilian region, which led to development of "Terra Preta de Indio" soils [69]. Biochar has long been used to date archaeological deposits due to its persistence in the environment [70]. Within the past decade, biochar has been evaluated as a potential alternative to nutrient releases and leaching reduction from the soil [16]. However, the standard application rate of biochar for specific soils and crop combination to obtain the maximum positive results is not available yet [71].

Biochar is the by-product of any type of biomass that has undergone pyrolysis (see example in Figure 2) [20,72]. Pyrolysis is a process that changes biomass to a carbon-rich-by-product as a result of the thermal degradation of organic materials by heating it to high temperatures in the absence of oxygen [70,73]. The pyrolysis process can be subdivided into separate categories: gasification (>800 °C), fast pyrolysis (~500 °C), and slow pyrolysis (450–650 °C) [74]. Slow pyrolysis is the best optimum pyrolysis process for the production of biochar [75,76]. The removal of volatile substances and the creation of crystalline carbons via condensations in biochar due to the increase in temperature from 400–500 °C enhance the adsorption abilities by generation of more pores [77,78]. Biochar is distinguishable from charcoal because of its usage as a soil amendment [21,79]. Responses of biochar are specific to the soil and climate within an area, biomass material, preparation method, and conditions [80,81]. Laird et al. [19] demonstrated that biochar carbon contents can range from <1% to >80% because of different biomass materials and pyrolysis conditions. Applied biochar in soils cannot be removed, so its use on a large scale has potential negative impacts on occupational health, environmental pollution, water quality, and food safety that need to be assessed [76].

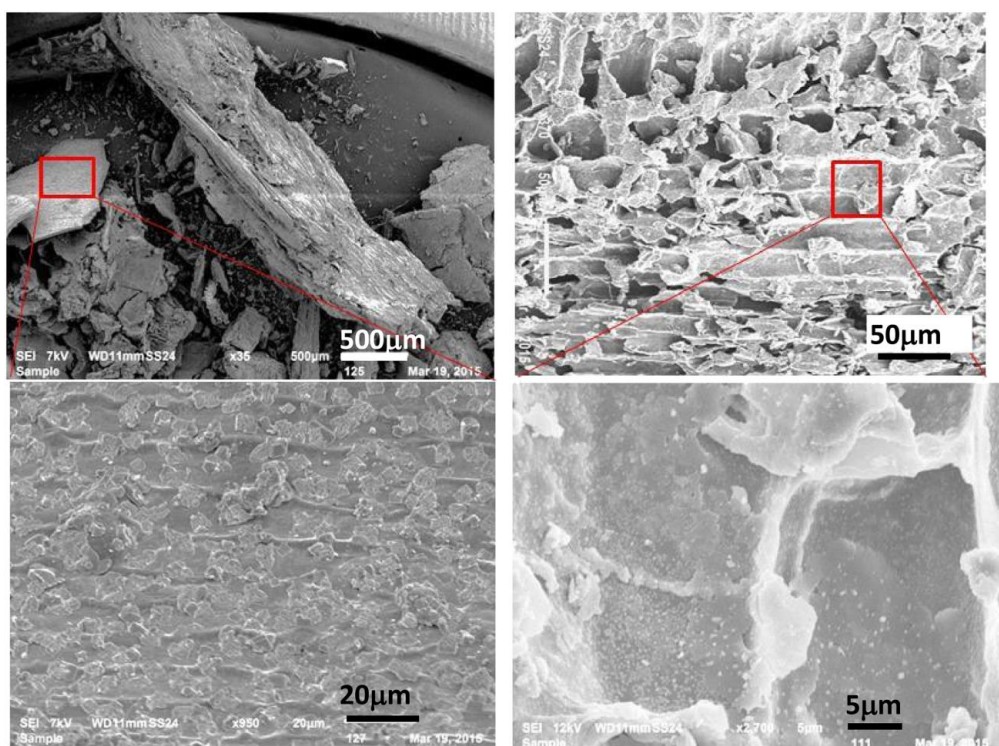

**Figure 2.** Scanning electron microscopy (SEM) pictures of pine sawdust biochar [72]. Reprinted from the reference with the permission, © 2021 John Wiley and Sons, Inc.

The char produced via pyrolysis is only known as biochar due to its amendment use in environmental management and production benefits to soil [18,73]. Bio-adsorbents similar to biochar are low cost with a high adsorption efficiency, as they require limited maintenance in wastewater contaminant treatments compared to other conventional methods [74–76]. The commercial worth of bio-adsorbents is low, and they are also accessible in abundance [76]. Affordability of an adsorbent can be increased, as they are stable and recyclable; hence, there is a high capacity for treatment of larger volumes of water contaminants over time [77]. Biochar's removal efficiencies for contaminants can be comparable to other commercial activated carbons because of improvements in cost-effective engineered biochar [78]. Biochar is also cheaper than other bio-adsorbents such as activated carbon, as it requires less production energy [74]. In addition to biochar's usage as a soil amendment, it is also used for carbon sequestration, mitigation of climate change, as a source of bio-energy, and waste management [18,70]. The high fraction of aromatic arrangements and high fraction of recalcitrant carbon (C) in biochar causes its resistance to chemical and biological degradation [82]. Biochar can persist in the soil for hundreds to thousands of years [70,83,84].

### 3.1. Properties of Biochar

Biochar characteristics such as the chemical composition, surface chemistry, particle and pore size distribution, and physical and chemical stabilization mechanisms in soils determine its effects on soil functions and faecal contaminants control [21]. Studies into biochar have demonstrated potentials for its use in increasing nutrient [19,70,85–88] and water retention [89–93] in soils, filtering heavy metals [94,95], reducing transport of microbes [14,15,22], increasing C sequestration [96–98], infiltration, soil aeration, root development, soil density, cation exchange capacity (CEC), and pH value [99–102]. The direct influence on soil structure, distribution of pore size, and density of the soil improves water holding capacity, aeration, and permeability [91,103,104].

Long-term properties including the stabilization of organic matter, slower release of nutrients from organic matter, and increased retention of cations have a huge impact to

reducing the contamination of water resources [104,105]. Adsorption mechanisms studies showed that various types of interactions such as chemical bonding, chemical interaction, (complexation and/or precipitation), physical adsorption, ion exchange, and electrostatic attraction are largely responsible for binding faecal wastewater contaminants [22,94,95]. Physical sorption of metallic contaminants occurs on the surface area and pore volumes of biochar due to the high affinity of adsorption retained within the pores [74,106]. Most positively charged contaminants are sorbed through electrostatic attractions, ligands specificity, and several functional groups (e.g., hydroxyl, Alternariol-AOH, carboxylate, ACOOH) on biochar because of their negatively charged surfaces [107]. The effect can also promote surface complexities and precipitation of these contaminates to their physical mineral phases, which immobilizes them [108]. Physical or surface sorption also happens by diffusional movement of organic and inorganic elements into sorbent pores [74]. Contaminant sorption also occurs through the exchange of ionizable cations or protons and chemical bonding on the biochar surface with those species in solution. Furthermore, biochar's high pH influences adsorption, because it affects charges on the surface, levels of ionization, and speciation of the adsorbent [74,108]. These characteristics of biochar make it a viable soil and water quality amendment in studies associated with on-site sanitation systems and agricultural sludge usage. However, the effect of the ageing process on biochar properties has not been studied in detail; for example, adsorption capacities of biochar change with time [71].

### 3.2. Biochar Usage in Faecal Sludge Management

#### 3.2.1. Nutrient Retention

Retention of soil nutrients has a direct effect to minimize risks of runoff and subsurface contamination of water bodies, highly reducing eutrophication and losses of nutrients [16]. Biochar can be a sustainable solution to latrine soil-bed nutrient leaching, consequently decreasing the nutrient concentrations in runoff and groundwater sources [16]. An increase in the CEC of a soil results in improved nutrient sorption on the colloids of biochar [19,20]. Biochar in soils also have the potential to largely decrease nitrogen losses and carbon dioxide releases [25]. Laird et al. [19] demonstrated an increase in N, organic C, P, K, Mg, and Ca in fine-loamy soil treated with hardwood biochar. A sorghum produced biochar also improved organic C and minimized greater losses of N, P, and K in overflow when combined with the soil [89]. Dissolved $NO_3$-N and $PO_4$-P decreased in wastewater bodies treated with a waste wood biochar-treated soil column [25] and also in soil mixed with an agricultural char (pecan, walnut, and coconut shells and rice hulls) [91]. Other than $NO_3$-N and $PO_4$-P, Beck et al. [92] observed a decrease in total N, total P, and total organic C. It has been seen that an increase in the application rates of biochar can also cause an increase in the nutrient holding capacity of the soil [20,90]. In their study, Huggins et al. [109] illustrated the efficiency of biochar to retain $NH_4^+$ and $PO_4^{3-}$ from faecal wastewater (Table 4).

**Table 4.** Wastewater treatment and the retention ability of biochar [109].

| Nutrient in Wastewater | Biochar Retention |
|---|---|
| $NH_4^+$ removal rate (g/m$^3$/d) | $5.4 \pm 0.51$ |
| $NH_4^+$ removal (%) | $90\% \pm 4\%$ |
| $PO_4^{3-}$ removal rate (g/m$^3$/d) | $3.8 \pm 0.01$ |
| $PO_4^{3-}$ removal | $87\% \pm 2\%$ |

#### 3.2.2. Heavy Metal Immobilization

Biochar can also act as a barrier to prevent heavy metals from percolating into groundwater aquifers and surface water resources [15,24,25]. Biochar has a high ability of filtering of heavy metals in contaminated soil and faecal sludge due to the potential of adsorbing metals on its surface [16]. Sequestration of Pb, Cd, Cu, and Ni has been reported from cottonseed hull biochar because of functional groups on the biochar surfaces [24]. As pH,

volatile matter, O:C, and N:C ratios in the biochar increase, biochar's capacity to adsorb heavy metals also increases [16,24]. In a study using poultry litter biochar and green waste biochar, it was found that Cd and Pb elements in soil water decreased [94]. Conversely, Cu increased in the soil water because of more mobility through increased dissolved organic C [94]. Other studies observed no effect on Cu when the soil was mixed with a hardwood biochar produced at 750 °C [20]. Decreases in Cd, Cr, Cu, Pb, Ni, and Zn in excess faecal effluent leachates from a soil column amended with a wood-based biochar were also found [25]. Cu, Cd, and Pb were removed from aqueous solution after amending the soil with bamboo, sugarcane, hickory, and peanut hull biochars, with the bamboo biochar being most effective [62]. The high adsorption of heavy metals in the study by Zhou et al. [62] was associated with pH increases in the solution. An increase in Cu and Zn removal as pH increased has also been seen using hardwood and corn straw biochar produced at temperatures of 450 °C and 600 °C [110]. These findings are similar to Krueger et al. [111], showing the immobilization occurring when faecal sludge is treated with biochar (Figure 3). The long-term impacts of biochar on heavy metal elements sorption are limited and require more research due to the recalcitrance of biochar [16].

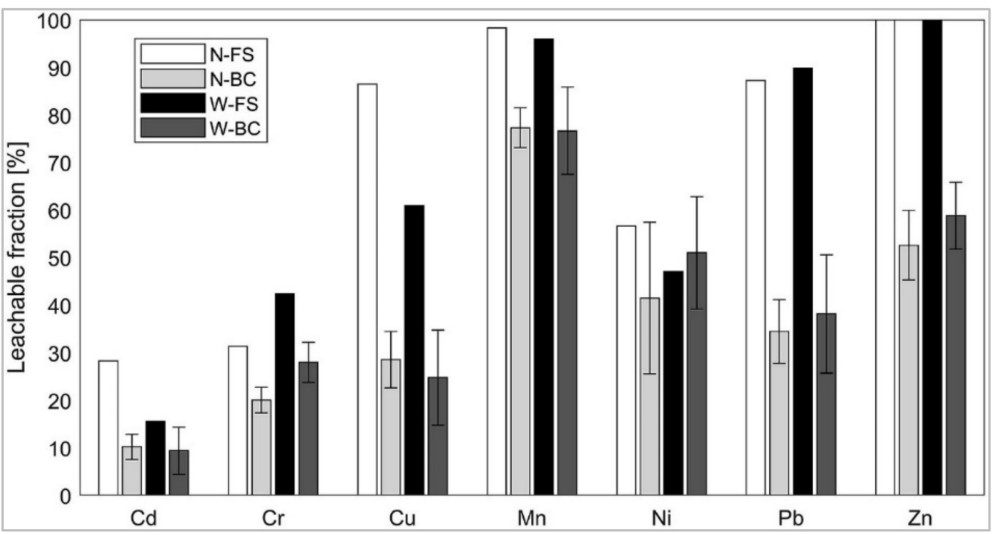

**Figure 3.** Mobility of heavy metals in faecal sludge (FS) and their derived biochars (BC); sludge sourced from Narsapur (N-FS) and Warangal (W-FS) Faecal Sludge Treatment Plants [111]. Reprinted from the reference with the permission, © 2019 Krueger et al., CC-BY-4.0.

### 3.2.3. Microbial Transport

The detection of *E. coli* and faecal coliforms (>1 CFU/100 mL) in soil and water resources above the guideline threshold [37–39] have a high risk on human health [16]. Presence of these bacteria indicates recent pollution from a faecal source such as pit latrine sanitations [13]. Bacteria can be infiltrated through the soil towards groundwater or move through overflow across the surface [16]. This threat to public health has urged investigation into microbial migration in soils, for which biochar amendments may be a solution [14,15].

When amended with soil, biochar can raise soil pH, which is essential for the mitigation of micro-organic pathogens such as *E. coli* and faecal coliform bacteria [22]. An increase in the organic matter, pH, conductivity, and dissolved organic C in a sandy soil using poultry manure biochar resulted in decreased soil *E. coli* and faecal coliforms migration [13]. Individually, these soil properties have been related to bacterial transport through soil [112–114]. Bolster and Abit [14] also demonstrated that biochar application rate, pyrolysis temperature, and *E. coli* surface properties largely contribute to the likely soil migrations. The higher temperature biochar (700 °C) exhibited a larger decrease in pathogen transport, possibly owing to the reduced negative surface charge of high-temperature biochars. The

improved surface area of high-temperature biochars provides a higher adhesion of *E. coli* cells [15]. Biochar can also assist in dehydrating excreta because of its high water holding capacity, reducing malodour by adsorption [23] and thereby helping to keep insects such as flies away.

Biomass type used for biochar pyrolysis also plays a role in the transportation of soil *E. coli*, and faecal coliforms [16]. Comparison between poultry litter and pine chip biochars indicated that the internal pore structure of the woody biochar retained or adsorbed more bacteria [15]. Additionally, soils with higher clay contents have fewer detachments because of the electrostatic attraction force of the negatively charged microbes and the positive clay functional groups [15]. The influence of biochar on microbial movement through soil relies on biomass material, temperature, and soil texture [16]. Results from literature on the effect of biochar use in faecal sludge and contaminant reduction are presented in Table 5.

**Table 5.** Comparative literature values from previous faecal wastewater and sludge studies using biochar.

| Parameter | | Type of Biochar | Concentration before Treatment | Concentration after Treatment | Literature |
|---|---|---|---|---|---|
| Nutrients | $NH_3$-N (mg/L) | Sludge and yellow pine biochar | 2.8 | 3.0 | Williams [16] |
| | $NO_3$-N (mg/L) | Sludge and yellow pine biochar | 0.6 | 1.5 | Williams [16] |
| | Nitrate (mg/L) | Effluent and waste wood pellets biochar | 27 | 3.0 | Reddy et al. [25] |
| | $NH_4$ (mg/L) | Wastewater and lodge pole pine wood biochar | 50 | 5 | Huggins et al. [109] |
| | Phosphate (mg/L) | Effluent and waste wood pellets biochar | 0.57 | 0.4 | Reddy et al. [25] |
| | Phosphate (mg/L) | Wastewater and lodge pole pine wood biochar | 18 | 2 | Huggins et al. [109] |
| | Phosphate (mg/L) | Faecal sludge and biochar | 31 | 6.2 | Krueger et al. [111] |
| Bacteria | *E. coli* (MPN/100 mL) | Effluent and waste wood pellets biochar | 7400 | 5000 | Reddy et al. [25] |
| | *E. coli* (MPN/100 mL) | Waste effluent and Monterey pine + eucalyptus biochar | 291 | <1 | Kranner et al. [115] |
| | *E. coli* (CFU/100 mL) | Effluent and poultry litter and pine chips biochar | 87 | 1.6 | Abit et al. [15] |
| | Faecal coliforms (MPN/100mL) | Sludge and yellow pine biochar | 150 | 26 | Williams [16] |
| | Enterococci (MPN/100mL) | Waste effluent and Monterey pine + eucalyptus biochar | 146 | 1 | Kranner et al. [115] |
| Heavy metals | Arsenic (mg/L) | Wastewater and lodge pole pine wood biochar | 27.9 | 0.01 | Huggins et al. [109] |
| | Cadmium (mg/L) | Effluent and waste wood pellets biochar | 24 | 17 | Reddy et al. [25] |
| | Cadmium (ppm) | Aqueous concentrations + bamboo, sugarcane bagasse, hickory wood, and peanut hull biochars | 30 | <1 | Zhou et al. [62] |
| | Cadmium (mg/L) | Wastewater and lodge pole pine wood biochar | 11.1 | <1 | Huggins et al. [109] |
| | Cadmium (mg/L) | Faecal sludge and biochar | 13.5 | 1.2 | Krueger et al. [111] |
| | Chromium (mg/L) | Effluent and waste wood pellets biochar | 5.13 | 5 | Reddy et al. [25] |
| | Chromium (mg/L) | Wastewater and lodge pole pine wood biochar | 34 | 0.1 | Huggins et al. [109] |
| | Chromium (mg/L) | Faecal sludge and biochar | 56.1 | 19 | Krueger et al. [110] |
| | Copper (mg/L) | Effluent and waste wood pellets biochar | 5 | 0.12 | Reddy et al. [25] |
| | Copper (ppm) | Aqueous concentrations + bamboo, sugarcane bagasse, hickory wood, and peanut hull biochars | 30 | <1 | Zhou et al. [62] |
| | Copper (mg/L) | Wastewater and lodge pole pine wood biochar | 8.3 | 0.04 | Huggins et al. [109] |
| | Copper (mg/L) | Faecal sludge and biochar | 463 | 209 | Krueger et al. [111] |
| | Lead (mg/L) | Effluent and waste wood pellets biochar | 0.48 | <0 | Reddy et al. [25] |
| | Lead (ppm) | Aqueous concentrations + bamboo, sugarcane bagasse, hickory wood, and peanut hull biochars | 50 | <1 | Zhou et al. [62] |
| | Lead (mg/L) | Wastewater and lodge pole pine wood biochar | 13.5 | <1 | Huggins et al. [109] |
| | Nickel (mg/L) | Effluent and waste wood pellets biochar | 110.61 | 80 | Reddy et al. [25] |
| | Zinc (mg/L) | Effluent and waste wood pellets biochar | 0.86 | 0.5 | Reddy et al. [25] |
| | Zinc (mg/L) | Wastewater and lodge pole pine wood biochar | 90.4 | 0.03 | Huggins et al. [109] |
| Malodour | Malodour reconstitution solution (ORS) (O.U./$m^3$) | Malodour solution + bamboo char, faecal char, and pine char | 173 | 73 | Stetina [23] |
| | ORS+$H_2S$ (O.U./$m^3$) | Malodour solution + bamboo char, faecal char, and pine char | 181 | 49 | Stetina [23] |
| | Butyric acid (O.U./$m^3$) | Malodour solution + bamboo char, faecal char, and pine char | 15 | 7 | Starkenmann et al. [70]; Stetina [23] |
| | Indole (O.U./$m^3$) | Malodour solution + bamboo char, faecal char, and pine char | 23 | 12 | Stetina [23] |
| | p-Cresol (O.U./$m^3$) | Malodour solution + bamboo char, faecal char, and pine char | 15 | 7 | Starkenmann et al. [70]; Stetina [23] |

## 4. Challenges, Sustainability, and Potential in Application of Biochar in Sludge Management

The most common challenge within communities using pit latrines is the ethical norm on the acceptance to repurpose biochar-treated faecal sludge for crop production [17]. Even though the biochar-treated sludge by-product can have acceptable threshold levels for most heavy metals and inorganic and organic contaminants, societies and communities in most developing countries treat human sludge as undesirable waste. In addition, the lack of knowledge in the biochar production process remains a challenge. The International Biochar Initiative [73] set guidelines on standards for production of biochar for use as soil amendments. However, information on biochar production for use in the treatment of contaminates and faecal sludge is limited. Moreover, most communities using latrines have livestock, which relies on the biomass material also needed to produce biochar. Nonetheless, the use of biochar in pit latrine sludge management can also be made sustainable, as the production process of biochar is regarded as an efficient management method to dispose of many organic wastes. However, advantages and disadvantages between the economic cost (production) and benefit value (application) of biochar need to be carefully measured. In addition, to enhance economic availability, easier production processes and cheaper sources of raw biomass materials need to be discovered to enhance economic availability [116]. Heavy metals can contaminate faecal sludge if toilets are also used to dispose of materials other than faecal sludge [52]. Studies have reported that for any new technology to be successfully integrated in a society, community awareness and engagement is important [13,17,24].

Education on the appropriate use of toilets is important [52], and application of biochar in latrines can be viable since a typical standard pit latrine only measures an approximate pit area of 2 m × 2 m [31,42,61]. In comparison to uses for amendment purposes in soil fertility and agriculture, sludge treatment can be more cost-effective, as the required biochar quantities are less bulky. The use of biochar has also been proven to increase faecal sludge decomposition, which can reduce the pit filling rates and increase the lifespan of a latrine. Biochar is also now commercially produced, which can also increase accessibility for sludge treatment and management uses. The high adsorption properties of biochar for water pollutants can assist with in situ sorbent and faecal sludge treatments. Such low-cost adsorbents can improve water quality through contaminant management.

## 5. Conclusions

This review focused on the potential uses of biochar in faecal sludge management (FSM) practices in most developing countries relying on pit latrine sanitation systems. Initially, the designs of pit latrines and the potential ways pollutants may migrate towards water resources without biochar amendments were outlined from previous literature. To understand the pollutant pit latrine leaching threat, the composition (heavy metals and inorganic and organic contaminates) of the stored in situ faecal excreta is important. Possible ways were explored on the effectiveness of biochar use in aqueous waste contaminant adsorption. The physical and chemical properties of biochar mostly determine its adsorption ability as an adsorbent in faecal waste management. Potential challenges in the adoption of biochar in faecal sludge management were also reviewed. Motivation can also be necessary to encourage communities using latrines to adopt biochar as an alternative low-cost carbon-rich absorbent for faecal sludge treatment. Biochar has high potential to effectively treat faecal sludge and control the migration of pit latrine pollutants.

*Future Research Perspectives in Faecal Sludge Management*

Studies on characterization and potential applications of biochar for several uses have been performed and research gaps have been indicated. Potential research lines that can be summarized as: (i) focus on potential secondary ecological risks in the process of biochar production by screening and pre-treating raw material to remove pollutants derived from biomass [100]; (ii) long-term stability and effect of biochar on agricultural

soil characteristics; (iii) assessment of multifunctional biochar materials on multi-heavy metals contaminated soils and as engineering application; (iv) cost-benefit analysis to enhance economic availability to improve production efficiency and reduce economic constraints [63,100]; (v) soil toxicity to organisms and plants induced by biochar [63]. These research gaps are also applicable for biochar use in faecal sludge management. The integration of biochar on faecal sludge management has the potential to be adopted by smallholder farmers who have limited access to fertilizers due to financial limitations. The success for this integration requires additional detailed studies considering that the smallholder farmers have more possibilities to produce charcoal than biochar:

- Similarities on biochar and charcoal application on faecal sludge management;
- Assess the effectiveness of production and use of biochar (or charcoal) as low-cost faecal treatment techniques to contain pollutants (i.e., heavy metals and microbial elements) based on dominant plant species to resolve the large variations in environments and respective mechanism;
- Assess the potential use of faecal sludge treated with biochar as soil amendment for nutrients sources and the risk to release heavy metals in agricultural production;
- The long-term impacts of biochar (or charcoal) recalcitrance in faecal sludge on heavy metal elements retention and nutrient release in agricultural production;
- Socio-economic benefits from use of faecal sludge amended with biochar as soil fertilizer for agricultural production;
- Potential retention of pollutants by biochar and/or charcoal from hydraulic loads in dry pit latrines;

**Author Contributions:** Conceptualization, M.M., J.J.v.T. and M.P.A.; methodology, M.M., J.J.v.T., M.P.A., A.B.J.C.N., L.M.C. and A.C.; validation, J.J.v.T., M.P.A., A.B.J.C.N., L.M.C., A.C., H.C., J.C.M. and R.B.A.R.; investigation, M.M., J.J.v.T., M.P.A., A.B.J.C.N., L.M.C., A.C., H.C., J.C.M. and R.B.A.R.; resources, J.J.v.T., L.M.C. and A.C.; data curation, M.M.; writing—original draft preparation, M.M.; writing—review and editing, M.M., J.J.v.T., M.P.A., A.B.J.C.N., L.M.C., A.C., H.C., J.C.M. and R.B.A.R.; supervision, J.J.v.T. and M.P.A.; project administration, J.J.v.T., L.M.C. and A.C.; funding acquisition, J.J.v.T., L.M.C. and A.C. All authors have read and agreed to the published version of the manuscript.

**Funding:** This research was funded by the National Research Fund (NRF) and South Africa–Mozambique–Zambia NRF Trilateral Joint Research (ZAM180911357528-118479).

**Institutional Review Board Statement:** The study was conducted according to the guidelines of the Declaration of Helsinki, and approved by the General/Human Research Ethics Committee (GHREC)-N.o-UFSHSD2019/1012. Environmental and Biosafety Research Ethics Committee (EBREC)-N.o-UFS-ESD2019/0066, University of the Free State, SA.

**Informed Consent Statement:** Informed consent was obtained from all subjects involved in the study. Written informed consent has been obtained from the patient(s) to publish this paper.

**Conflicts of Interest:** The authors declare no conflict of interest.

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
