# Peer review of "Potential Use of Biochar in Pit Latrines as a Faecal Sludge Management Strategy to Reduce Water Resource Contaminations: A Review"

_applsci, doi:10.3390/app112411772_

Round 1

Reviewer 1 Report

More than 50% of World population ( about 4.2 billion people) do not have access to safe sanitation, and most of the urban poor, namely in Sub-Saharan counties rely on latrines and feacal sludge services .M

The topic of the manuscript is considered relevant, clear and with a lot of references.

Some editorial suggestions:

pp1, line 36-37- present the keywords by alphabetic order;

pp2,3,  line 96-98- As on-site sanitation systems, it should also be considered: ecological latrine , fossa alterna, anaerobic biogas reactor and septic tank.

pp5, concerning "Heavy metal composition of sludge", some reference should be done to the risks of contamination from  sludge from industrial sources.

pp5, line 170-  In most developing counties there are no pit latrines and dry sanitation any more, but septic tanks with an adequate final disposal of the effluent- Typically sludge is dried on site (on drying beds) or discharged on existing WWTP for further treatment. So, please correct the sentence " In most of developing countries, filled up latrines are either closed or.....".

Apparently references  96, 97 and 98 seem to be missing along the text.

Author Response

Please find the attached report

Reviewer 2 Report

The paper contributes to the issue. Just is necessary recent references in the introduction.

And use standard unit, please, revise all mg/l ou mg/L or ppm. In the manuscript and in the table.

The author must improve the biochar low-cost adsorbent examples.

The authors must improve the mechanism of chemical bonding, chemical interaction,  physical adsorption, ion-exchange, and electrostatic attraction are largely responsible for binding faecal wastewater contaminants.

Author Response

Please find the attached report
